# When Small Models Team Up: A Weak-Expert Ensemble Surpassing LLMs for Automated Intellectual-Property Audits

## Abstract

Intellectual Property Rights (IPR) enforcement on e-commerce platforms is increasingly challenged by the scale and complexity of modern online marketplaces, where counterfeit goods and brand infringements evolve rapidly to evade detection. Existing approaches rely heavily on human review or unimodal AI models, limiting their scalability and adaptability. We introduce IPR-GPT, a multimodal system for automated IPR auditing that combines a Multi-Agent Audit Framework with an Audit Reversal data augmentation mechanism. MAAF leverages a structured multi-agent collaboration strategy, combining specialized experts in visual analysis, textual reasoning, legal compliance, and exemption handling. By systematically exploring the trade-offs between Vision-Language Models and lightweight vision models, MAAF achieves a performance boost of up to 24.26% in key IPR audit tasks. AR further enhances model robustness by synthesizing hard-to-classify audit cases, improving generalization in real-world scenarios. Extensive experiments demonstrate that IPR-GPT consistently outperforms state-of-the-art models, setting a new benchmark for multimodal IPR enforcement. Our results challenge the prevailing belief that **larger multimodal LLMs are always superior**, showing instead that a purpose-built ensemble of weak experts can deliver both higher accuracy and lower cost.

## 1 Introduction

The rapid growth of e-commerce platforms has led to an unprecedented increase in online product listings, many of which pose IPR risks, including counterfeit goods, unauthorized replicas, and brand misrepresentation(Kwok et al., 2004; Singh, 2023). These violations not only compromise brand integrity but also undermine consumer trust. Traditional IPR enforcement primarily relies on manual review processes, where human auditors inspect product listings to identify non-compliant items. However, this approach is inherently inefficient, prone to inconsistencies, and unable to scale with the sheer volume of daily product uploads, leading to gaps in enforcement and increased regulatory challenges (Hendrycks et al., 2021).

Existing research on IPR auditing has focused predominantly on unimodal solutions, where either textual or visual data is analyzed in isolation. Text-based methods, such as BERT-based classifiers, detect brand infringement through product descriptions and metadata analysis (Devlin et al., 2019; Lu et al., 2019). And image-based techniques employ convolutional neural networks and Vision Transformers to identify counterfeit products based on visual characteristics (Radford et al., 2021; Tan & Bansal, 2019; Wang et al., 2021). While these methods have improved automation, they struggle to address multimodal violations, where textual descriptions and images are intentionally manipulated to evade detection. Moreover, these approaches lack adaptability to the rapidly evolving landscape of online product listings, limiting their effectiveness in real-time enforcement.

With recent advancements in vision-language models, many e-commerce platforms have begun integrating VLMs into their compliance pipelines, leveraging their multimodal capabilities for automated auditing (Chen et al., 2022; AI, 2025; OpenAI, 2024). In our experiments, VLM-based pipelines underperform a simple hybrid design that combines long chain-of-thought reasoning with lightweight embedding-based classifiers. The hybrid system is cheaper to run and more accurate on

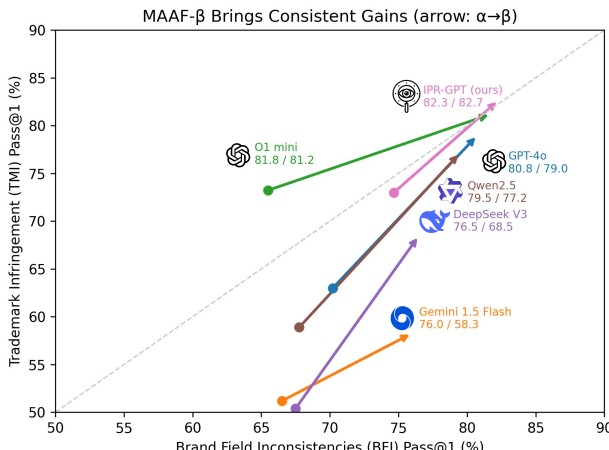

Figure 1: A comparative analysis of the performance of various LLMs and VLMs in IPR Audit Benchmark with MAAF-$\alpha$ and $\beta$, including BFI pass@1 and TMI pass@1 accuracy.

IPR-Audit, suggesting that current VLMs are still less reliable than text-only LLMs for this type of decision-making.

To systematically analyze this issue, we introduce IPR-Audit, a dataset comprising 1,837 diverse product listings collected from the E-Commerce platform over the past year. This dataset captures two of the most prevalent IPR violations: Brand Field Inconsistency (BFI) and Trademark Infringement (TMI), which together account for nearly half of all reported IPR cases. **Based on IPR-Audit, we propose a Multi-Agent Audit Framework (MAAF), consisting of specialized experts, including an Image Expert, an Information Expert, an Exemption Expert, an IPR Rules Expert, and a Comprehensive Audit Expert.** Each agent operates independently, leveraging either a well-engineered LLM prompt, a dedicated classification model, or a lightweight embedding-based system to ensure efficient and accurate decision-making (Shen et al., 2023; Gibney, 2025).

- Weak-Expert Ensemble Framework(MAAF-$\beta$). We decompose IPR auditing into complementary sub-tasks, logo spotting, NER-based brand mention extraction, multimodal cross-checking, etc., and assign each to a lightweight model, achieving performance with up to 24.26% improvements and reducing GPU memory beyond 60%.

- Audit Reversal data augmentation for robustness. We introduce the AR, a technique that systematically synthesizes high-quality training data by inverting expert decisions, generating additional audit samples that help the model learn from ambiguous or borderline cases.

- Composition over concatenation for IPR auditing. We propose IPR-GPT, a large-scale model tailored for intellectual property audits, and show that combining it with MAAF outperforms simply feeding all inputs into existing state-of-the-art models.

Our key takeaway is that **Combining hand-crafted weak models, each specializing in a narrowly scoped signal channel, yields quantifiably better and cheaper performance than stuffing all inputs into one large multimodal LLM.**

## 2 RELATED WORK

### 2.1 IPR ENFORCEMENT

Research on IPR enforcement in e-commerce has predominantly focused on automated systems to detect counterfeit goods, unauthorized usage, and trademark violations. Early approaches were unimodal, relying solely on either textual or visual data, limiting their capacity to address complex multimodal IPR infringements effectively.Text-based methods employ NLP models such as BERT to classify product descriptions and identify brand violations through semantic analysis (Hu et al., 2023). However, they struggle against adversarial manipulations where product descriptions are

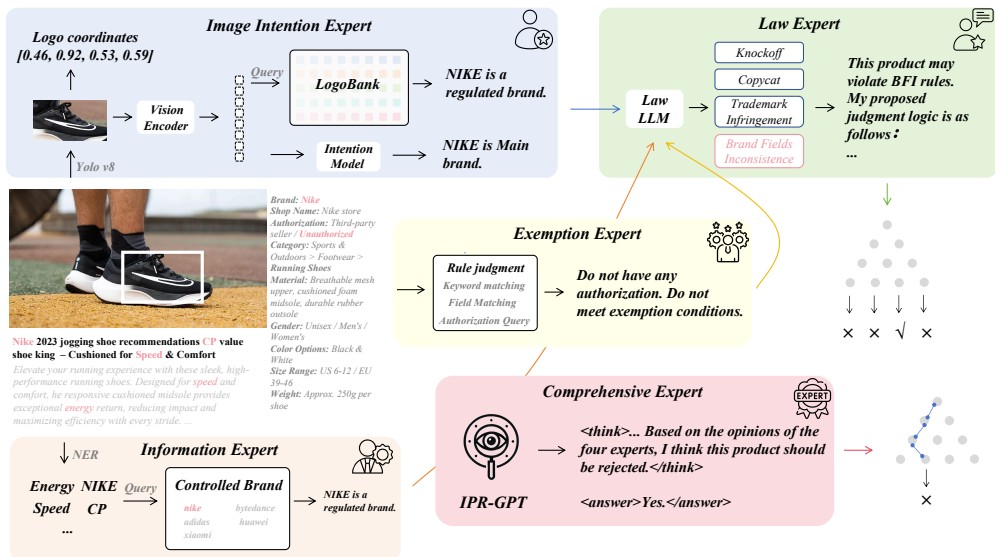

Figure 2: Pipeline of MAAF-$\beta$ for IPR enforcement.

deliberately altered to evade detection. Image-based approaches leverage deep learning models, including CNNs and ViTs, to analyze product images for counterfeit detection(Chen et al., 2021; Tursun et al., 2019). Despite their effectiveness, these models remain vulnerable to sophisticated evasion tactics, where counterfeiters modify both visual and textual elements to bypass detection(Trappey et al., 2021; Alshowaish et al., 2022).

To overcome these challenges, MAAF integrates fast embedding models, NER-based text analysis, and Long CoT reasoning, enabling a multimodal, real-time auditing system. The image intent auditing expert in MAAF combines a visual embedding model, an intent classification model. This structured, multi-agent approach ensures a comprehensive and adaptable IPR auditing framework, outperforming unimodal and end-to-end models in real-world enforcement scenarios.

## 2.2 VISION-LANGUAGE MODELS

Vision-Language Models have rapidly advanced, with models such as LLaVA, Qwen-2 VL, and InternVL demonstrating strong multimodal reasoning capabilities(Team, 2024; Liu et al., 2024; Chen et al., 2024). These models integrate text and image processing within a unified framework, excelling in tasks like image captioning, visual question answering, and multimodal retrieval. Foundational models like CLIP and ALIGN leverage contrastive learning to align visual and textual representations(Radford et al., 2021; Li et al., 2021), significantly improving multimodal classification and retrieval performance. Because these advancements, many e-commerce platforms have begun integrating VLMs for IPR auditing, exploring their potential for automated compliance(Hendrycks et al., 2021). However, our findings suggest that VLMs still underperform compared to hybrid approaches combining Long CoT reasoning with embedding-based classifiers.

In contrast, MAAF employs a structured multi-agent approach, leveraging specialized unimodal agents instead of a monolithic VLM. The image intent expert combines a visual embedding model, an intent classification model, and LogoBank. This structured design enables MAAF to achieve scalable, real-time auditing with greater adaptability and interpretability, outperforming general-purpose multimodal models in dynamic e-commerce environments.

## 3 METHODOLOGY

Due to the lack of prior work on IPR, possibly because business data and methods need to remain confidential, it has been difficult for us to conduct comparisons with other methods. Therefore, we have designed two self-comparison frameworks, MAAF-$\alpha$ and MAAF-$\beta$.

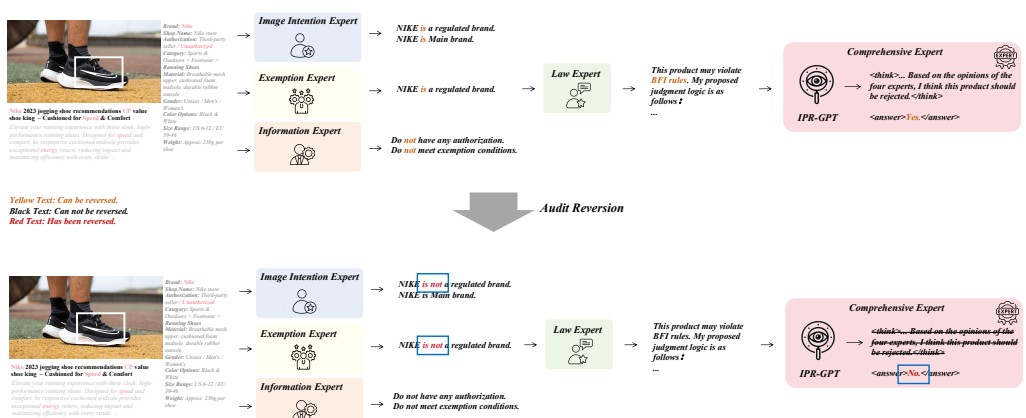

Figure 3: Illustration of the Audit Reversal (AR) mechanism applied in MAAF for IPR enforcement. Where yellow denotes reversible elements, black represents non-reversible parts, and red indicates successfully reversed elements.

### 3.1 MULTI-AGENT AUDIT FRAMEWORK

#### 3.1.1 OVERALL ARCHITECTURE

The MAAF framework integrates specialized experts for IPR compliance, implemented in two versions to compare VLMs with traditional vision-based models. MAAF-$\alpha$ employs LLMs and VLMs for autonomous text and image processing, serving as a benchmark for assessing the feasibility of multimodal generative models in IPR auditing. We also propose MAAF-$\beta$, which replaces full LLM inference with structured combinations of lightweight embedding models, intent classifiers, and rule-driven logic. Each expert in MAAF-$\beta$ operates independently, leveraging precomputed embeddings and specialized classifiers rather than relying solely on generative models.

The following sections focus on MAAF-$\beta$, detailing its expert components and demonstrating its superior accuracy with reduced computational costs. And MAAF-$\alpha$ is implemented using VLM.

#### 3.1.2 IMAGE INTENTION EXPERT: IMAGE AND INTENTION RECOGNITION

This expert evaluates brand presence in product images to determine potential IPR violations. Using YOLOv8 as the base model, fine-tuned on a collected logo dataset, the system systematically identifies brand elements.Each detected logo is encoded using the ViT-L/14-336 visual encoder from UNICOM(An et al., 2023), fine-tuned on the LogoDet 3K and Logo-2k+ dataset(Wang et al., 2024; 2022; 2020) and matched against LogoBank, a curated embedding database of brand logos, using cosine similarity:

$$\mathbf{l}_{\text{sim}} = \underset{\mathbf{l}' \in \mathcal{L}}{\arg\max} \frac{\mathbf{l} \cdot \mathbf{l}'}{\|\mathbf{l}\| \|\mathbf{l}'\|} \tag{1}$$

If the highest similarity score surpasses a threshold, the brand is assigned to the detected logo. The model further classifies brand appearance into five categories—background, co-branded, subbranded, group-branded, or main product branding—via a 1.8B-parameter Valley classifier (Wu et al., 2025).

#### 3.1.3 INFORMATION EXPERT: NER-BASED BRAND RECOGNITION

The Information Expert extracts brand mentions from textual data and verifies them against a predefined controlled brand list. Given a product description $t$, a BERT-based Named Entity Recognition (NER) model first detects potential brand mentions. Each detected brand is then matched against Controlled Brand List:

$$b_{\text{match}} = \mathbb{I}(b \in \mathcal{C}) \tag{2}$$

where $\mathcal{C}$ denotes the controlled brand set, and $\mathbb{I}(\cdot)$ is an indicator function returning 1 if a match is found and 0 otherwise.

The final output consists of verified brand mentions:

$$E_{\text{text}} = \{b \in B_t \mid b_{\text{match}} = 1\} \tag{3}$$

Where $B_t$ represents the brand identified by BERT-based NER model given a product description $t$.

### 3.1.4 EXEMPTION EXPERT: EXEMPTION IDENTIFICATION

The Exemption Expert verifies IPR exemptions using rule-based logic and metadata, covering categories like second-hand goods, co-branded items, group brands, and specific industries such as collectibles or toys.

Given a product's textual and categorical information, the exemption verification process begins by extracting relevant metadata fields, such as authorization status, product category, and seller type. Let $A$ represent the extracted authorization details, and let $C$ denote the product category. The exemption decision function is defined as:

$$E_{\text{exm}} = \text{match}(A, C, \mathcal{D}_{\text{auth}}) \tag{4}$$

where $\mathcal{D}_{\text{auth}}$ is a structured database containing predefined exemption rules. The verification process consists of two key stages:

First, if a product is sold by an officially authorized distributor or brand-affiliated entity, it is eligible for exemption. This is determined by checking whether $A$ appears in the set of verified authorizations as $A \in \mathcal{D}_{\text{auth}}^{\text{verified}}$.

Second, certain product categories, such as second-hand marketplaces or authorized co-branded items, qualify for exemption regardless of authorization status. This is evaluated as $C \in \mathcal{D}_{\text{auth}}^{\text{category}}$.

If either condition is satisfied, the product is exempted from IPR violation assessment. Otherwise, it is forwarded to subsequent experts for further scrutiny. To enhance the precision of exemption verification, this expert also applies field-level consistency checks, ensuring that product descriptions, images, and metadata align with the claimed exemption type.

### 3.1.5 LAW EXPERT: MULTI-RULE EVALUATION MODEL

The Law Expert determines IPR violations by selecting the appropriate legal rule based on outputs from the Information, Image Intention, and Exemption Experts. Unlike conventional systems that assess only one type of violation, this expert dynamically adapts to various legal violations based on product attributes.

Given a product $P$, the system aggregates relevant evidence and The Law Expert then selects the most relevant legal framework:

$$E_{\text{law}} = f(E_{\text{info}}, E_{\text{img}}, E_{\text{exm}}), \quad R_{\text{law}} = \text{select}(\mathcal{R}, E_{\text{law}}) \tag{5}$$

where $f$ represents the reasoning function integrating textual, visual, and exemption-related attributes, $\mathcal{R}$ is a predefined set of audit rules. A 1.8B-parameter Valley model classifies the product's violation type, ensuring consistent legal assessments.

### 3.1.6 COMPREHENSIVE EXPERT: REASONING-BASED DECISION MAKING

The Comprehensive Expert synthesizes outputs from all prior experts to reach a final audit decision. Each contributes specialized insights: $E_{\text{info}}$ identifies brand details, $E_{\text{img}}$ assesses detected logos and their intent, $E_{\text{exm}}$ verifies exemptions, and $E_{\text{law}}$ provides legal classifications. Comprehensive

Expert is based on a chain-of-thought LLM or a multi-agent inference engine, such as DeepSeek R1 or IPR-GPT. The model produces a verdict of either $\boxed{Reject}$ or $\boxed{Approve}$, with potential extensions for more granular decisions depending on platform policies.

## 3.2 AUDIT REVERSAL DATA AUGMENTATION

Audit Reversal (AR) enhances IPR audit robustness by systematically modifying expert outputs to generate adversarial training cases. Unlike conventional data augmentation, AR keeps the input product information unchanged while reversing expert decisions, forcing the reasoning model to generalize across ambiguous cases and reducing misclassification rates.

Given a product $P$, the original audit decision $D_{\text{orig}}$ is determined by the outputs of the Information, Image Intention, Exemption, and Law Experts:

$$D_{\text{orig}} = \text{reason}(E_{\text{info}}, E_{\text{img}}, E_{\text{exm}}, E_{\text{law}}). \tag{6}$$

To generate adversarial samples, AR reverses one or more expert outputs while keeping the product description and image unchanged:

$$D_{\text{rev}} = \text{reason}(\text{reverse}(E_{\text{info}}, E_{\text{img}}, E_{\text{exm}}, E_{\text{law}})). \tag{7}$$

If $D_{\text{rev}} \neq D_{\text{orig}}$, the case is successfully reversed, and both versions are included in the training dataset.

Each expert's output is reversed systematically to generate edge cases. Audit Reversal systematically modifies expert outputs to generate adversarial training samples. The reversal process includes altering both textual and visual brand recognition, causing the audit decision to shift accordingly. Similarly, flipping the Exemption Expert's output changes whether a product qualifies for exemption, directly influencing the audit outcome. Lastly, modifying the Law Expert's classification reassigns the violation category, ensuring the reasoning model adapts to varying legal interpretations. By applying these structured reversals, AR enhances model robustness and adaptability in complex auditing scenarios.

## 3.3 IPR-GPT TRAINING

IPR-GPT is a large-scale model fine-tuned for IPR auditing, based on the Qwen2.5-72B base model. Training consists of supervised fine-tuning (SFT) with structured audit data from MAAF and reinforcement learning (RL) with Audit Reversal data to enhance decision robustness.

SFT is performed on 1.4k IPR-Audit cases, where IPR-GPT is optimized to reproduce the Comprehensive Expert's reasoning and final decision rather than individual expert outputs. Each training sample is structured as $(X, \mathcal{T}, D)$. where $X$ contains product attributes, $\mathcal{T}$ is the chain-of-thought (CoT) reasoning, and $D$ is the final audit decision. The CoT reasoning follows:

$$\mathcal{T} = \text{CoT}(E_{\text{info}}, E_{\text{img}}, E_{\text{exm}}, E_{\text{law}}). \tag{8}$$

Following SFT, RL is applied using 3k adversarial training samples generated by Audit Reversal. Unlike SFT, which focuses on reasoning consistency, RL optimizes for final decision correctness using a reward function:

$$R = \begin{cases} +1, & \text{if answer} = \text{gold answer} \\ -1, & \text{otherwise.} \end{cases} \tag{9}$$

RL is trained via proximal policy optimization (PPO) with reward-adjusted loss. By reinforcing accurate decision-making under adversarial conditions, RL prevents overfitting to expert outputs. The final model acts as the reasoning engine for the Comprehensive Expert. At inference, it synthesizes expert evaluations into a structured reasoning trace and final decision.

| Model | Modality | Parameters | Year | Performance(pass@1) | | |
|---|---|---|---|---|---|---|
| | | | | **BFI** | **TMI** | **Average** |
| GPT-4o | Text | - | 2024 | 70.21% | 62.99% | 66.60% |
| Gemini 1.5 Flash | Text | - | 2024 | 66.50% | 51.18% | 58.84% |
| Gemini 1.5 Flash | T+I | - | 2024 | 69.00% | 62.99% | 66.00% |
| O1 mini | Text | - | 2024 | 65.50% | **73.23%** | **69.37%** |
| DeepSeek R1 | Text | 671B | 2025 | **70.50%** | 61.42% | 65.96% |
| DeepSeek V3 | Text | 671B | 2024 | 67.50% | 50.39% | 58.95% |
| QvQ | T+I | 72B | 2024 | 70.00% | 59.06% | 64.53% |
| Qwen2 VL | T+I | 72B | 2024 | 60.50% | 45.67% | 53.09% |
| Qwen2.5 | Text | 72B | 2024 | 67.76% | 58.88% | 63.32% |
| *with MAAF-$\beta$* | | | | | | |
| GPT-4o | Text | - | 2024 | 80.76%(+10.55%) | 79.00%(+16.01%) | 80.12%(+13.52%) |
| Gemini 1.5 Flash | Text | - | 2024 | 76.00%(+9.50%) | 58.27%(+7.09%) | 67.14%(+8.30%) |
| O1 mini | Text | - | 2024 | **81.76%(+24.26%)** | 81.2%(+12.97%) | **81.48%(+18.62%)** |
| DeepSeek R1 | Text | 671B | 2025 | 75.00%(+4.50%) | **84.25%(+22.83%)** | 79.63%(+13.67%) |
| DeepSeek V3 | Text | 671B | 2024 | 76.50%(+9.00%) | 68.50%(+18.11%) | 72.50%(+13.56%) |
| Qwen2.5 | Text | 72B | 2024 | 79.50%(+16.74%) | 77.17%(+18.29%) | 78.34%(+17.52%) |

Table 1: Comparison of IPR audit performance across various models w/wo the MAAF-$\beta$.

## 4 EXPERIMENTS

In this section, we evaluate the performance of our proposed MAAF-$\beta$, AR and IPR-GPT across multiple experiments. We begin by describing the training environment, datasets, and evaluation metrics. Because the elimination of specific experts will lead to the loss of overall framework functionality, we only compare them with baseline models, perform ablation studies, and analyze the real-time adaptability.

### 4.1 EXPERIMENTAL SETUP

All experiments were conducted on a distributed system with 32 NVIDIA GeForce H100-SXM-80GB GPUs. The base model used for IPR-GPT is Qwen2.5-72B Instruction, which was fine-tuned in two stages. In the first stage, supervised fine-tuning was applied with an initial learning rate of $1e^{-4}$, weight decay of $5e^{-4}$, a warmup ratio of 0.01, 3 epochs and cosine annealing for learning rate decay. The batch size was set to 8 with a gradient accumulation of 2 steps. In the second stage, reinforcement learning was introduced to optimize the model's decision-making. The training data was collected from 2024 e-commerce marketplace and spans multiple product categories, including apparel, electronics, and household goods. Each sample contains complete product information provided by sellers, such as product titles, descriptions, specifications, and images. The dataset consists of a training set of 1606 samples from products listed between January and July 2024 and a test set of 277 samples from newly listed products between August 1 and August 10, 2024. All samples were annotated by domain experts for compliance and non-compliance with intellectual property regulations.

To evaluate model performance, three key metrics were used. BFI pass@1 measures the accuracy of detecting brand field inconsistencies, TMI pass@1 evaluates the ability to identify trademark infringement cases, and Average pass@1 computes the overall model performance by averaging BFI and TMI accuracy. For the performance of unimodal models in MAAF-$\alpha$, we use the Gemini 1.5 Flash model as the MLLM for the Image Intention Expert, while the remaining experts utilize unimodal models.

### 4.2 PROJECTED RESOURCE FOOTPRINT

Before any code was executed we built a first-principles hardware budget for the two configurations. In MAAF-$\alpha$ every expert stage uses a 72B vision-language model. Storing those weights in FP16 precision consumes roughly 144 gigabytes of device memory and the activations created during a forward pass raise the instantaneous footprint to about one hundred sixty gigabytes. On an H100-80GB accelerator that footprint forces frequent tensor swapping and yields an amortised inference time of around 2.1 seconds per listing when the batch size is one. In MAAF-$\beta$ the early vision and text specialists stay lightweight, yet the final reasoning expert is also a 72B model. The live parameter count therefore reaches about 74 billion, occupying close to one hundred fifty gigabytes

| Model | Performance (pass@1) | | |
|-------|------|------|---------|
| | **BFI** | **TMI** | **Average** |
| IPR-GPT (MAAF-$\beta$) | **82.31%** | **82.67%** | **82.49%** |
| IPR-GPT (MAAF-$\alpha$) | 74.66% | 72.98% | 73.32% |
| IPR-GPT (w/o MAAF) | 65.49% | 60.13% | 62.81% |
| IPR-GPT (w/o AR) | 79.12% | 78.10% | 78.51% |
| Qwen2.5 | 67.76% | 58.88% | 63.32% |

Table 2: Ablation study of IPR-GPT performance.

in FP16 precision. Because earlier stages pre-compute their embeddings offline, the peak memory seen during live inference falls to roughly 75 gigabytes and latency drops to about 0.8 seconds. This gives an estimated two-times reduction in memory demand and roughly a two-and-a-half-times speed-up relative to MAAF-$\alpha$. Subsequent measurements aligned with these projections within 20 percent, confirming the usefulness of the estimate for capacity planning.

## 4.3 MAAF FRAMEWORK PERFORMANCE COMPARISON

We evaluate MAAF-$\beta$ with a range of models, including GPT-4o, Gemini 1.5 Flash, DeepSeek R1, DeepSeek V3, Qwen2 VL, and others. The results show that adding MAAF-$\beta$ leads to significant improvements across all three metrics. Notably, models that originally operate on text-only inputs, such as DeepSeek R1, experience substantial gains, with BFI pass@1 increasing from 70.50% to 75.00% and TMI pass@1 improving from 61.42% to 84.25%. Similarly, text+image models such as Qwen2.5 exhibit strong improvements, with BFI pass@1 rising from 67.76% to 77.17% and TMI pass@1 increasing from 58.88% to 77.17%.

The observed performance gains indicate that MAAF-$\beta$ effectively enhances IPR auditing by integrating specialized expert modules for textual and visual analysis. Compared to standalone VLMs, MAAF-$\beta$ offers improved interpretability and adaptability, ensuring more accurate and consistent decisions in detecting IPR violations.

## 4.4 COMPARISON OF VISION-LANGUAGE MODELS AND DEDICATED VISION MODELS

With the increasing adoption of VLMs in e-commerce platforms, their effectiveness in IPR auditing remains a subject of investigation. To evaluate this, we compare VLM-based implementations MAAF-$\alpha$ with traditional vision models within the MAAF framework MAAF-$\beta$. In the baseline MAAF-$\beta$ setup, the Image Expert consists of VIT L/14-336 as the vision encoder, a fine-tuned YOLOv8 for logo detection, and a Tiny Valley-based intent classifier. We contrast this with MAAF-$\alpha$, where the Image Expert is replaced with various VLMs such as DeepSeek V3, Gemini 1.5 Flash, and GPT-4o, leveraging their built-in multimodal reasoning capabilities. As presented in Table 3, models utilizing VLMs for image analysis consistently underperform compared to those employing dedicated vision models.

The performance gap is evident in the BFI and TMI pass@1 metrics. DeepSeek V3, when used as the Image Expert, achieves a BFI pass@1 of 63.93% and a TMI pass@1 of 61.42%, significantly lower than traditional vision-based approaches. Similarly, Gemini 1.5 Flash struggles with fine-grained logo detection, leading to lower scores across all evaluated metrics. While GPT-4o demonstrates relatively strong results, it still falls short of the structured vision models used in MAAF-$\beta$. The highest-performing approach remains IPR-GPT(MAAF-$\beta$), which achieves 82.31% in BFI and 82.67% in TMI, **surpassing all VLM-based implementations.**

## 4.5 IPR-GPT PERFORMANCE EVALUATION

We evaluate the effectiveness of IPR-GPT integrated with MAAF-$\beta$ by comparing its performance against multiple baseline models in the task of IPR violation detection. The evaluation considers BFI pass@1, TMI pass@1, and Average pass@1 metrics across models such as DeepSeek R1, DeepSeek V3, Gemini 1.5 Flash, GPT-4o, O1 Mini, and Qwen2.5. As shown in Table 4, IPR-GPT(MAAF-$\beta$) **achieves the highest performance across all key metrics,** demonstrating its superior capability in automated IPR enforcement. Notably, compared to the next best-performing model, O1 Mini, which achieves 81.76% in BFI pass@1 and 81.2% in TMI pass@1, IPR-GPT(MAAF-$\beta$) demonstrates a measurable improvement in classification accuracy.

| Model | Performance(pass@1) | | |
|---|---|---|---|
| | BFI | TMI | Average |
| DeepSeek R1 | 81.82% | 71.67% | 76.75% |
| DeepSeek V3 | 63.93% | 61.42% | 62.68% |
| Gemini 1.5 Flash | 67.00% | 66.57% | 66.67% |
| GPT-4o | 86.27% | 76.15% | 81.21% |
| Qwen2vl | 72.73% | 68.33% | 70.53% |
| QvQ | 74.58% | 75.76% | 75.17% |
| **MAAF-$\beta$** | 82.31% | **82.67%** | **82.49%** |

Table 3: Comparison of IPR-GPT(MAAF-$\beta$) against other models with MAAF-$\alpha$. The table presents BFI pass@1, TMI pass@1, and Average pass@1.

| Model | Performance (pass@1) | | |
|---|---|---|---|
| | BFI | TMI | Average |
| DeepSeek R1 | 75.00% | 84.25% | 79.63% |
| DeepSeek V3 | 76.50% | 68.50% | 72.50% |
| Gemini 1.5 Flash | 76.00% | 58.27% | 67.14% |
| GPT-4o | 80.76% | 79.00% | 80.12% |
| O1 Mini | 81.76% | 81.2% | 81.48% |
| Qwen2.5 | 79.50% | 77.17% | 78.34% |
| **MAAF-$\beta$** | 82.31% | 82.67% | **82.49%** |

Table 4: Comparison of IPR-GPT (MAAF-$\beta$) against other models with MAAF-$\beta$. The table presents BFI pass@1, TMI pass@1, and Average pass@1.

### 4.6 ABLATION STUDY

To assess the impact of different components in IPR-GPT, we conduct an ablation study by systematically removing key elements and evaluating their effects on performance. We analyze the model's performance under four settings: the full IPR-GPT with MAAF-$\beta$, IPR-GPT with MAAF-$\alpha$, IPR-GPT without MAAF, and IPR-GPT without Audit Reversal (AR) augmentation. The results are presented in Table 2.

The results show that IPR-GPT (MAAF-$\beta$) achieves the highest performance across all metrics, with a BFI pass@1 of 82.31%, TMI pass@1 of 82.67%, and an Average pass@1 of 82.49%. In contrast, when using MAAF-$\alpha$, which replaces structured experts with VLM-based reasoning, performance drops significantly to 74.66% in BFI and 72.98% in TMI, leading to an overall decline of 9.17% in the Average pass@1 score. **This demonstrates that while MAAF-$\alpha$ offers a fully LLM-driven approach, it is less effective than MAAF-$\beta$ in IPR enforcement.**

Further removing MAAF entirely results in even greater performance degradation, with BFI pass@1 decreasing to 65.49% and TMI pass@1 dropping to 60.13%, reducing the overall accuracy to 62.81%. This suggests that the structured multi-agent auditing framework is crucial for enhancing the model's decision-making capability. Additionally, removing AR data augmentation causes a noticeable performance decline, with a 3.98% reduction in BFI pass@1 and a 4.57% drop in TMI pass@1 compared to the full IPR-GPT (MAAF-$\beta$). **This confirms that AR contributes significantly to model robustness by providing diverse and challenging training cases.**

## 5 CONCLUSION

In this paper, we introduced IPR-GPT, a novel framework for automated intellectual property rights (IPR) enforcement in e-commerce. By combining the Multi-Agent Audit Framework (MAAF) with Audit Reversal Data Augmentation (AR), we created a robust system that integrates multimodal inputs for detecting IPR violations, including brand field inconsistencies and trademark infringements. We presented a new task and dataset, IPR-Audit, comprising diverse product data from e-commerce platforms, which serves as a benchmark for evaluating IPR enforcement models. Our approach significantly outperforms existing models, demonstrating the effectiveness of MAAF and AR in improving accuracy and adaptability in real-world auditing. Furthermore, the modular nature of our framework allows for easy extension to other compliance tasks beyond IPR, providing a versatile solution for automated auditing in various domains. **These findings overturn the widespread assumption that ever-larger multimodal LLMs are inherently superior, demonstrating instead that a carefully orchestrated ensemble of lightweight, task-focused experts can achieve greater accuracy at a fraction of the computational cost.**

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

## A    ETHICS STATEMENT

This research adheres to ethical standards in data collection, annotation, and usage. All product listings used in the IPR-Audit dataset were obtained from publicly accessible e-commerce platforms and anonymized to remove any personally identifiable information, seller identities, and sensitive business data. The study strictly focuses on automated intellectual property auditing for compliance research and does not target specific merchants or brands. The dataset construction process followed responsible data handling principles, ensuring no violation of intellectual property rights or privacy regulations.

## B    REPRODUCIBILITY STATEMENT

We have made every effort to ensure the reproducibility of our experiments. All model architectures, hyperparameters, and training configurations are described in detail in the methodology section. The dataset construction procedure, expert annotation process, and evaluation metrics are explicitly defined to enable replication. Upon publication, we plan to release a reproducible implementation of the Multi-Agent Audit Framework (MAAF) and anonymized sample cases from the IPR-Audit dataset to facilitate future research on multimodal compliance auditing.

## C    THE USE OF LLMS

We did not use LLMs for research ideation, modeling decisions, or data analysis. Their only role was during manuscript preparation, where we used LLM-based tools to spot typos and minor grammar issues. The design of MAAF, the implementation of IPR-GPT, the construction of IPR-Audit, and all experiments and analyses were carried out by the authors.

