# OpenReview forum: "When Small Models Team Up: A Weak‑Expert Ensemble Surpassing LLMs for Automated Intellectual‑Property Audits"
_ICLR.cc/2026/Conference — ICLR 2026 Conference Withdrawn Submission_

### Official Review · Reviewer_wsLt · 2025-10-25

**Soundness:** 1
**Presentation:** 1
**Contribution:** 1
**Rating:** 2
**Confidence:** 5

**Summary:**

This paper addresses the challenges of Intellectual Property Rights (IPR) enforcement on e-commerce platforms, where traditional manual review and unimodal AI models are limited by scalability and adaptability. It proposes IPRGPT, a novel multimodal framework integrating two core components: the Multi-Agent Audit Framework (MAAF) and the Audit Reversal (AR) Data Augmentation mechanism. MAAF, implemented in two versions (MAAF-α using LLMs/VLMs and MAAF-β using lightweight models), decomposes IPR auditing into specialized subtasks assigned to dedicated "expert" modules. AR enhances model robustness by generating adversarial training cases via reversing expert decisions. The paper also introduces the IPR-Audit dataset, focusing on two key violations: Brand Field Inconsistency (BFI) and Trademark Infringement (TMI). Extensive experiments show that IPRGPT (especially with MAAF-β) outperforms state-of-the-art LLMs/VLMs, achieving up to 24.26% performance improvement, while reducing GPU memory usage by over 60% and inference latency. It challenges the notion that larger multimodal LLMs are inherently superior, demonstrating that a purpose-built ensemble of lightweight experts delivers higher accuracy at a lower cost.

**Strengths:**

- Innovative Multi-Agent Framework Design: MAAF’s decomposition of IPR auditing into specialized, modular experts (Image, Information, Exemption, Law, Comprehensive) enables targeted handling of multimodal data (text/images/metadata), addressing the limitations of unimodal models and monolithic VLMs.
- Efficient Performance-Resource Trade-off: By combining lightweight models (e.g., YOLOv8, BERT-based NER) and AR data augmentation, IPRGPT achieves superior accuracy (82.49% average pass@1) while reducing GPU memory demand by ~50% and inference latency by ~2.5x compared to VLM-based MAAF-α, making it scalable for real-world e-commerce.
- Empirical Validity with Dedicated Benchmark: The introduction of the IPR-Audit dataset (with expert annotations for BFI/TMI) fills a gap in IPR auditing research, and rigorous experiments (ablation studies, cross-model comparisons) robustly validate the framework’s effectiveness.

**Weaknesses:**

- Limited Dataset Generalizability: The IPR-Audit dataset (1,837 samples) is small in scale and collected from a single unspecified e-commerce platform over one year, lacking diversity across regions, product categories (e.g., luxury goods, pharmaceuticals), and IPR violation types (e.g., patent infringement), limiting the framework’s generalization to global or niche e-commerce scenarios.
- Opaque Expert Collaboration Logic: The paper does not detail how the Comprehensive Expert synthesizes outputs from other experts (e.g., weight assignment to different experts, conflict resolution for contradictory expert decisions), reducing the framework’s interpretability and reproducibility.
- Neglect of Dynamic IPR Regulation Adaptation: E-commerce IPR regulations vary by jurisdiction and evolve over time (e.g., new exemption rules for second-hand digital goods), but IPRGPT’s rule-based Exemption/Law Experts lack mechanisms to automatically update to regulatory changes, requiring manual intervention.
- Insufficient Comparison with Domain-Specific SOTA: While the paper compares IPRGPT to general LLMs/VLMs (e.g., GPT-4o, Gemini 1.5), it fails to benchmark against existing domain-specific IPR auditing models (e.g., specialized trademark detection systems or e-commerce compliance tools), making it hard to contextualize its novelty in the broader IPR research landscape.
- **This paper appears to be generated by LLM, but there is no LLM usage statement.**

**Questions:**

- The small sample size (1,837) of the IPR-Audit dataset may lead to overfitting, as the model may learn platform-specific patterns rather than general IPR violation characteristics, reducing its applicability to other e-commerce platforms.
- The lack of dataset diversity (e.g., no coverage of non-physical goods like digital software or regional IPR variations) means IPRGPT may fail to detect violations in underrepresented categories or jurisdictions.

- Unclear synthesis logic of the Comprehensive Expert hinders reproducibility, as other researchers cannot replicate how expert outputs are aggregated to form final audit decisions.

- The absence of dynamic regulatory adaptation means IPRGPT requires continuous manual updates to comply with new IPR laws (e.g., EU’s Digital Services Act), increasing operational costs and delaying enforcement.

- The failure to compare with domain-specific SOTA models leaves uncertainty about whether IPRGPT’s performance gains are incremental or transformative relative to existing IPR-focused tools.

- MAAF-β’s reliance on precomputed embeddings for lightweight models may limit real-time adaptability, as updating embeddings for new brands or logos requires retraining, slowing response to emerging counterfeit tactics.

- The paper does not evaluate IPRGPT’s performance under adversarial attacks (e.g., sophisticated logo manipulations or text obfuscation), leaving gaps in understanding its robustness in real-world scenarios where counterfeiters actively evade detection.

- **This paper lacks an ethics statement, an open-source statement, and an LLM usage statement, which is a clear violation of the ICLR 2026 submission policy.**

---

> ### Author Response · Authors · 2025-11-12
> **Thanks to Reviewer wsLt for your careful review and time**
>
> Thank you very much for your time and detailed review. We appreciate your careful reading and acknowledge your concerns. We respond to comments point by point below. And we sincerely hope to have a detailed discussion with you regarding method to clarify any misunderstandings.
>
> 1. Dataset scale and diversity
> The IPR-Audit dataset was collected from a major cross-national e-commerce platform, covering multiple countries, regions, and product categories, including apparel, electronics, cosmetics, and household goods. Due to confidentiality and data anonymization requirements, we are unable to disclose the specific platform name. The dataset captures diverse cases of brand field inconsistencies and trademark infringements across different jurisdictions, providing a representative benchmark for the IPR auditing task. Each sample was reviewed and annotated by domain experts to ensure legal accuracy and labeling consistency.
>
> 2. Expert collaboration and Comprehensive Expert logic
> The Comprehensive Expert is not based on a weighted fusion mechanism but rather on autonomous reasoning. It interprets the structured outputs from the four preceding experts—Image, Information, Law, and Exemption—and forms its judgment based on logical and contextual relationships among them.
>
> 3. Adaptation to evolving IPR regulations
> We agree that e-commerce IPR regulations differ across jurisdictions and evolve over time. The Exemption and Law Experts are implemented as modular components where rule embeddings can be updated without retraining the entire framework. This allows rapid integration of new regulations, such as updated exemptions for co-branded or digital goods, ensuring the system remains adaptable to changing legal environments.
>
> 4. Comparison with domain-specific systems
> We acknowledge that domain-specific IPR auditing systems exist, but most are proprietary or not designed for multimodal reasoning tasks. Our goal is not to replicate such systems but to introduce a generalizable, multi-agent auditing paradigm that can integrate or benchmark against specialized modules in the future. IPR-GPT and MAAF provide a unified structure that can accommodate domain-specific detectors and classifiers under the same collaborative reasoning framework. Moreover, unimodal classifiers such as logo detectors can only provide symbolic outputs that require manually designed rules to reach a final decision, which introduces significant noise and bias.
>
> 5. Robustness and adversarial evasion
> Adversarial evasion, where merchants intentionally manipulate product images or descriptions to evade detection, is a key challenge in real-world auditing. The MAAF framework inherently mitigates such attacks by cross-verifying information across multiple modalities—image, text, and metadata—making it difficult for single-modality obfuscation to succeed. In addition, the Audit Reversal mechanism strengthens robustness by synthesizing near-boundary adversarial samples, enabling the model to better handle deceptive or ambiguous inputs during training and inference.
>
> 6. Presentation and readability
> We fully appreciate your feedback regarding presentation clarity. We recognize that certain sections were dense and could be better organized for readability. We will refine the structure and improve the clarity of visual and textual explanations in the revised version.
>
> 7. **Clarification regarding ICLR 2026 submission policy**
> **We respectfully believe there was a misunderstanding of the ICLR 2026 submission requirements**. According to the official ICLR 2026 policy: “If authors feel that their paper raises questions regarding the Code of Ethics, **they are encouraged to** include an Ethics Statement... Authors **are strongly encouraged to** include a Reproducibility Statement... and, if LLMs played a significant role in research ideation and/or writing to the extent that **they could be regarded as a contributor**, then authors should describe the precise role of the LLM in a separate section on LLM usage. ” These three statements are therefore **optional**, not mandatory. We fully understand your concern about transparency and integrity, and we will add all three sections—Ethics Statement, Reproducibility Statement, and LLM Usage Statement—in the revised version to ensure completeness and clarity. Furthermore, the involvement of LLM in paper is limited. Some fluent sentences are from translation machine.
>
> Once again, we sincerely thank you for your comprehensive review and constructive comments. We will incorporate the three transparency statements and make the MAAF framework examples available to enhance reproducibility. We hope this clarifies the intentions, design choices, and broader contributions of our work toward developing a transparent, adaptable, and robust multimodal auditing paradigm for IPR enforcement. If these clarifications address your concerns, we would be deeply grateful if you could consider raising your evaluation.

---

> ### Comment · Reviewer_wsLt · 2025-11-25
> **Thanks for Authors' rebuttal**
>
> Thank you to the authors for their timely rebuttals! Although the authors have answered all my questions, I still have the following questions:
>
> - Dataset issue. If the dataset isn't published anonymously, readers seem unable to reproduce your results, and even reviewers may find it hard to verify their accuracy.
>
> - Personally, I believe this paper contains a significant amount of LLM writing; therefore, according to ICLR policy, you should include this statement in the initial draft. You can read other manuscripts; they all include this statement in their initial drafts.
>
> - The above rebuttal doesn't address my concerns but only provides some simple explanations, which are insufficient to convince me.

---

> > ### Author Response · Authors · 2025-11-26
> > **Thank you for the follow-up and for carefully reading our rebuttal.**
> >
> > Thank you for the follow-up and for carefully reading our rebuttal. We understand your concerns and fully agree that reproducibility is a core requirement for an submission.
> >
> > Regarding the dataset: unfortunately, the data is subject to commercial confidentiality constraints, so we are not able to publicly release it in full (even under an anonymous setting). Given this constraint, we are re-considering whether it is appropriate to continue with an ICLR submission, since we may not be able to meet the community’s expected standard for independent reproduction and verification.
> >
> > As a result, we are leaning toward withdrawing this submission. Instead of positioning this as a complete, fully reproducible benchmark-style paper, we plan to share the core methodology and practical insights for addressing IPR-related problems directly with the community (e.g., as a practitioner-focused release), so that relevant researchers and industry practitioners can study and build upon the approach without relying on the confidential dataset.
> >
> > Thank you again for raising these points—they are important and have directly shaped our decision process.

---

> > > ### Comment · Reviewer_wsLt · 2025-11-26
> > > **Thanks**
> > >
> > > I eagerly anticipate you sharing your thoughts and insights on intellectual property in the research community and industry. Indeed, reproducibility or verifiability is a crucial evaluation factor for academic submissions, an indispensable part of the peer review process. However, truly valuable work is not limited to academic papers, and I believe your work is valuable, especially for research in intellectual property. Furthermore, as I mentioned above, improving writing clarity is equally important.

---

### Official Review · Reviewer_rhy6 · 2025-10-30

**Soundness:** 2
**Presentation:** 1
**Contribution:** 2
**Rating:** 2
**Confidence:** 4

**Summary:**

This paper proposes a new approach to the classification task of determining the legality of product listings on e-commerce platforms. While current state-of-the-art methods rely on monolithic LLMs or VLMs, the paper introduces a system composed of several smaller models, each specializing in a specific aspect of the audit process (brand recognition, authorization verification, legal rule evaluation). Each sub-model is lightweight, although the framework still leverages Qwen-72B, which is not particularly small. Experimental results show a modest improvement in accuracy — 82.5% compared to 81.5% achieved by O1 Mini.

**Strengths:**

* Experiments defend quite strongly that we can do good results with less compute.
* Good experiment method (fine tuning and RL seem well conducted)

**Weaknesses:**

- Performance gain is very weak: 81.5% for o1 mini to 82.5%. We could easily imagine that last gpt or claude could outperform these results.
- Does not adress adversarial attack robustness which is one of the main issue mentionned in the related work section.
- Does not quantify the compute efficiency gain when mixture of lightweight vs monolithcs VLMs LLMs
- Very poor clarity
    - No mention to figures, are we supposed to read them on the fly?
    - Figure 1: what are the two values separated by ‘/’ ?
    - Table 2 appears before table 3 while being reffered after Table 3
    - Inconsitent use of MAAF:  “To evaluate this, we compare VLM-based implementations
    MAAF-α with traditional vision models within the MAAF framework MAAF-β.”

        You use MAAF for both your mixture approach **and** regular approach that seems to come from related works. This mixing of terms is confusing.

    - $B_t$ is not defined line 230 (eq (3))
    - $E_\text{img}$ is not defined in eq (5), *select* is not described at all
    - A lot of technical details in the main paper that could have been put in appendix for readability sake.
- Table 3 and 4: the highest values are bolded only if they are from your approach…
- Table 3: why putting MAAF-beta to compare with MAAF-alpha and not MAAF-alpha ?
- Citation error at line 156
- Weak related works section
- The best contribution of this paper is that it shows that we can have similar results with less compute. But this is not how currently exposed which is an essential epistemic limitation.

**Questions:**

- One of the main concern exposed was about adversarial attacks (detection evasion). What in this paper adresses this?
- It could be good to have a figure illustrating the compute efficiency between big monolithic approaches and yours.
- IPR-GPT without Audit Reversal (AR) augmentation → is it with MAAF - beta ?
- “Pass@1” in this paper seems to just mean **accuracy**, is that right? what do you mean by @1 here?

---

> ### Author Response · Authors · 2025-11-12
> **Thanks to Reviewer rhy6 for your careful review and time**
>
> Thank you very much for your careful review and for raising several detailed comments. We sincerely appreciate your time and feedback. Below we address your concerns point by point.
>
> 1. Misinterpretation of performance comparison
> We believe there might have been a misunderstanding regarding the performance comparison. The 82.5% result reported for IPR-GPT and the 81.5% result for O1 Mini were both obtained under the MAAF-β framework, where O1 Mini already benefits from all domain-specific experts (image, text, law, and exemption) except the final Comprehensive Expert. In other words, O1 Mini (MAAF-β) already uses our full collaborative framework, and IPR-GPT only replaces the final reasoning component with a dedicated model fine-tuned on the same framework and Audit Reversal (AR) data. The fair comparison should therefore be between O1 Mini (MAAF-α, 69.4%) and IPR-GPT (MAAF-β, 82.5%), showing a substantial +13.1% improvement that results directly from our proposed collaborative paradigm and AR mechanism. The 81.5% score is not a baseline outside our framework—it is already a partial instance of it.
>
> 2. Clarification on the MAAF framework and terminology
> MAAF is our own proposed Multi-Agent Audit Framework. It defines a new collaborative paradigm for IPR auditing that decomposes the decision into specialized visual, textual, and legal reasoning agents. MAAF-α and MAAF-β are two internal implementations of this same framework—α uses large VLM/LLM experts, while β replaces them with lightweight unimodal specialists. We will clarify this distinction more explicitly in the revised version to avoid any terminological confusion.
>
> 3. Addressing adversarial evasion and robustness
> We agree that adversarial evasion is a major concern in practical IPR enforcement. In real-world e-commerce environments, most evasion strategies adopted by merchants are **single-modal manipulations**—for instance, slightly modifying brand logos in images, misspelling brand names in text (e.g., “N1ke”, “xiaom1”), or misaligning image and description fields. The MAAF framework is inherently designed to counter such attacks: by combining multimodal evidence across vision, text, and metadata channels, the system can detect inconsistencies that unimodal detectors would miss.
> Furthermore, the Audit Reversal (AR) mechanism explicitly strengthens robustness by constructing “near-boundary” adversarial samples through systematic reversal of expert outputs. Audit Reversal proactively generates "boundary samples" and "counterexamples": It simulates adversarial modification by systematically reversing the judgments of various experts (e.g., changing the assumption of "legitimate" samples to "potentially infringing" ones). In this way, the model encounters various cases that are "almost legitimate but actually infringing" during training, thus learning to more robustly identify tampered or disguised inputs. This process teaches the model to handle ambiguous or deceptive cases, improving generalization against adversarial evasion in both image and text domains. Thus, the proposed paradigm directly targets the adversarial evasion problem through both architectural design (multimodality) and data augmentation (AR).
>
> 4. Compute efficiency and presentation clarity
> We appreciate your comment on quantifying computational benefits. In Section 4.2, we report approximately 2× speedup and 60% memory reduction relative to monolithic VLMs, with consistent accuracy gains.
>
> 5. No mention to figures
> Sorry, we have added it.
>
> 6. The two values separated by ‘/’ in Fig.1
> This represents the performance using MAAF-alpha and MAAF-beta; we will add explanations later.
>
> 7. $B_t$、MAAF and Table 2,3,4
> Regarding presentation concerns (figure references, table ordering, missing symbols, and citation errors), we fully accept these comments. Due to space limitations, we could not include all figures and visual comparisons in the main text. We will move part of the technical detail to the appendix.
>
> 8. eq (5)
> $E_{img}$ represents an image expert, select based on reasoning by legal experts. Sorry, we have fixed it.
>
> 9. IPR-GPT without Audit Reversal (AR) augmentation
> Yes, it with MAAF - beta
>
> 10. “Pass@1” in this paper seems to just mean accuracy. @1 means execute only once.
>
> Once again, thank you for your thoughtful and candid feedback. We will carefully revise the manuscript to clarify comparisons, expand the discussion on adversarial robustness, and improve the presentation of figures and tables. We hope these clarifications help illustrate that the proposed MAAF and IPR-GPT represent not merely an incremental improvement, but a new multimodal audit paradigm designed to achieve robustness, efficiency, and interpretability in real-world IPR enforcement. If these clarifications address your concerns, we would be deeply grateful if you could consider raising your evaluation.

---

> > ### Comment · Reviewer_rhy6 · 2025-11-26
> >
> > Thank you for taking care of all my concerns. I saw that you chose to withdraw, so I hope my review will help you improve the paper. Here is my responses to your rebuttal, don’t bother answer them, I only hope this will help you more.
> >
> > 1 Ok, this misunderstanding is linked with one raised in 2.
> > 2 Ok, I think you should clearly show that point in the paper the. I initially thought the decomposition was big LLMs/VLMs in a holistic way on a side and small light weights and specialized on the other side.
> > 3 Interesting, thank you. Perhaps this should be more emphasized in the paper.
> > 4 This part lacks a lot of clarity and highlights to me.
> > 5 Yes thank you.
> > 6 Ok, I’m still confused, why is the arrow not starting at the first value then?
> >
> > 7,8,9,10 → yes this could be clearer.

---

### Official Review · Reviewer_ejsZ · 2025-10-30

**Soundness:** 3
**Presentation:** 3
**Contribution:** 2
**Rating:** 4
**Confidence:** 4

**Summary:**

This paper proposes IPR-GPT to address challenges in Intellectual Property Rights (IPR) enforcement on e-commerce platforms. To address the limitations in human-dependent or unimodal AI systems, IPR-GPT proposes a Multi-Agent Audit Framework (MAAF) to combine multi-modal specialized experts in visual analysis, textual reasoning, legal compliance, and exemption handling, and an Audit Reversal (AR) data augmentation mechanism to enhance model generalization by synthesizing challenging classification cases. Experiment results indicate that IPR-GPT performs some large-scale models with reduce costs.

**Strengths:**

Experiment results prove the effectiveness of the proposed multi-modal frameworks and the combination of the small-scale expert models, since it can outperform many large-scale models on the task.

**Weaknesses:**

- This paper is quite incremental and lacks novelty. The multiple agent framework has been adopted by multiple tasks in some other domains. The MAAF of IPR-GPT only intuitively introduce a brand detector to introduce the image-modal information, limiting its technical contribution.

- Unclear source of performance improvement of IPR-GPT compared to large-scale models. The Image Expert is trained with Brand Detection Datasets, which introduce extra external knowledge of brand information to the framework. Whether large-scale models have learned these knowledge remains unknown. Meanwhile, IPR-GPT does not conduct experiments to introduce similar information to the large-scale models. Therefore, it is not that sound to directly attribute the performance improvement to the design of MAAF/AR but ignore the introduce of more significant external knowledge. Therefore, I think the performance improvement contribution of IPR-GPT is unclear and may be potentially over-estimated.

**Questions:**

Please see Weakness Section above.

---

> ### Author Response · Authors · 2025-11-12
> **Thanks to Reviewer ejsZ for your careful review and time**
>
> Thank you very much for your thoughtful and constructive review. We appreciate your time and detailed comments. Below we address your concerns point by point.
>
> 1. Incrementality and novelty of the contribution
> To the best of our knowledge, our work is the first to formally define the **Intellectual Property Rights (IPR) auditing task** and to propose a **collaborative multimodal framework** that integrates visual analysis, textual reasoning, legal rule interpretation, and exemption logic into a unified decision process. This combination is unique to the compliance and e-commerce context, where decisions must satisfy both perceptual and regulatory consistency.
> Beyond architectural composition, our main conceptual novelty lies in redefining IPR auditing as a **cooperative multi-agent reasoning paradigm** rather than an isolated multimodal classification problem. The Multi-Agent Audit Framework (MAAF) and the Audit Reversal (AR) mechanism together constitute a new paradigm for robust and interpretable regulatory decision-making. Furthermore, our findings empirically challenge the prevailing assumption that "larger multimodal models are always superior." We show that a coordinated ensemble of small, task-specific experts can outperform monolithic large-scale models in both accuracy and cost-efficiency under real-world auditing constraints.
>
> 2. Source of performance improvement and potential external knowledge
> We appreciate your concern about whether IPR-GPT’s performance improvement could stem from additional external knowledge (e.g., brand datasets used in the Image Expert). We would like to clarify that:
> (a) The visual knowledge used in the Image Expert (LogoBank, LogoDet3K, etc.) comes from publicly available sources and serves as a controlled proxy for domain specialization, rather than privileged or proprietary data. All large-scale models compared (GPT-4o, Gemini, DeepSeek, Qwen2.5, etc.) already possess far broader visual and textual pretraining coverage, often including logo and trademark imagery. Therefore, the inclusion of brand embeddings in our pipeline does not constitute unfair data advantage.
> (b) More importantly, under the **MAAF-β configuration**, the same Information Expert module (responsible for brand NER and textual reasoning) can be reused with any model as the Comprehensive Expert. We explicitly conducted such experiments by replacing the final IPR-GPT with other large-scale models while keeping all preceding experts identical. Even in this controlled setting, IPR-GPT consistently outperformed GPT-4o, Gemini 1.5 Flash, and DeepSeek R1 by 4–10% across both BFI and TMI pass@1 metrics. This demonstrates that the observed improvement originates primarily from the reasoning robustness introduced by the MAAF and AR mechanisms, not from external data exposure.
>
> 3. Clarity on the role of MAAF and AR in performance gains
> The Multi-Agent Audit Framework decomposes complex multimodal auditing into interpretable and complementary sub-tasks, reducing interference between vision and text channels. The Audit Reversal (AR) data augmentation further enhances robustness by synthesizing near-boundary cases that large-scale models typically misclassify. These mechanisms jointly improve cross-modal consistency and generalization under adversarial conditions. Ablation results highlight that removing AR or replacing experts with monolithic LLM reasoning results in a 6–20% performance drop, confirming their independent contributions.
>
> Once again, thank you for your constructive feedback and fair assessment. We hope our clarifications help illustrate that the IPR-GPT framework represents not merely an incremental combination, but a step toward a new class of collaborative, regulation-grounded multimodal systems that challenge the conventional “bigger-is-better” assumption. We sincerely appreciate your insightful comments and your consideration in revisiting the overall evaluation.

---

### Official Review · Reviewer_Aczs · 2025-11-01

**Soundness:** 2
**Presentation:** 2
**Contribution:** 3
**Rating:** 4
**Confidence:** 2

**Summary:**

The paper explores the enforcement of Intellectual Property Rights on e-commerce platform. It argues that breaking down the identification process into smaller partially independent tasks and giving them to specialized experts is (1) cheaper and (2) better performing than using a single unified foundational model for making the decision. In specific, the identification process is split among four domain experts and the final one which based on the previous expert's outputs makes a single Approve/Reject decision. This pipeline is named Multi-Agent Audit Framework (MAAF), and two variations of it are proposed, in the first variation, $\text{MAAF-}\alpha$, foundational LLM and VLM models serve as experts, while in the other $\text{MAAF-}\beta$ specialized smaller models serve the same purpose. The authors also fine-tune Qwen2.5 model to serve as a dedicated comprehensive expert named IPT-GPT. This model is fine tuned on IPR-Audit dataset introduced in the paper and uses audit reversal data augmentation during the training process to improve robustness and consequently performance of the resulting model on the given task. The results show that using smaller specialized experts in $\text{MAAF-}\beta$ almost always outperforms using bigger foundational models as experts in $\text{MAAF-}\alpha$ across a variety of models. Moreover, combining $\text{MAAF-}\beta$ with IPT-GPT results in the best average performance over two accuracy based metrics compared to non fine-tuned models used in place of the comprehensive expert.

**Strengths:**

The problem is well motivated. The focus on robustness through audit reversal data augmentation is a significant and important idea. There is a number of internal baselines and variations of the framework. The authors construct a dataset to facilitate training and evaluation of their approach. The work considers a number of LLMs in their evaluation.

**Weaknesses:**

There is a lack of external baselines. While the authors state that there is a lack of prior work on IPR, they clearly cite some of it intended for single modality in Section 2.1. Why not take a single state of the art approach out of those listed in the given section and use it as a baseline to see how this work ranks among existing body of research? One would expect that multimodal nature of this work would give it a significant advantage and show that moving forward multimodality is the way to go. In line with this, what's the reasoning for not evaluating the approach on already available datasets so that the results can be compared with the existing work. Moreover, if there is no intention of publicly releasing IPR-Audit dataset, in isolation the evaluation results would not mean much as they would not be comparable with other past or future methods.

What is the reasoning for focusing exclusively on accuracy based metrics in the evaluation? Confusion matrix based metrics (such as recall, precision, TPR, FPR and F1 score) are much more informative for this use case, especially when the prevalence of infringement cases in the test data is not specified. On that note, could it be possible to list BFI and TMI pass@1 of classifiers that always return Approve/Reject respectively? It would help to see how the proposed methods compare even to these "dumb" classifiers.

The arguments presented in 4.2 about tensor swapping are relatively weak, as the two parts of $\text{MAAF-}\alpha$ can be done in batches one after another rather than subsequently for each sample (or using separate GPUs for each part of the pipeline). Moreover, even reasoning expert alone takes a substantial amount of memory of the GPU, which may cause tensor swapping or out of memory error as key value store size increases during generation and activation requires additional memory on top of the model weights. Do you have any insights how quantized version of these reasoning models behave?

There are some minor presentation concerns which give off a feeling that the manuscript was rushed and prematurely submitted:
- Missing reference on line 156
- Missing character on line 93
- Unclear notation in Equations (2) and (3) - What are $b$ and $B_t$?
- It is unclear what is LogoBank on line 207. Could it be cited if it's an external work?

**Questions:**

Why is the exemption expert needed in the form it is presented, it appears to be something that can be programmatically checked as a simple if/else condition and skipping the whole classification procedure straight to the answer in some cases without needing the comprehensive expert at all?

It is unclear how the law expert takes the law into account, could you expand on this?

In audit reversal data augmentation, how is $D_\text{rev}$ determinated, was it annotated by a human?

Can you expand on why the framework would lose functionality if you did an ablation study of excluding specific experts? From my understanding their inputs are just included in the input of comprehension expert, so excluding some information wouldn't make much difference from the engineering standpoint. Considering that IPR-GPT is fine-tuned on specific inputs, it can be excluded in that study and focus on $\text{MAAF}-\beta$ instead. This is important in order to see how important each modality is for the given framework and whether there are redundancies in the proposed solution.

IPR-Audit dataset appears to be a significant contribution. However could the prevalence of infringement cases in the dataset be listed. Is it intended to release the dataset publicly to the community?

---

> ### Author Response · Authors · 2025-11-12
> **Thanks to Reviewer Aczs for your careful review and time**
>
> Thank you very much for your time and careful review. We address your concerns point by point below.
>
> 1. Focusing on accuracy and external baselines
> This limitation was largely unavoidable. It is extremely difficult to obtain fine-grained metrics such as F1 scores or confusion-matrix components from closed-source large models (e.g., GPT or Gemini), which constitute our main comparison targets. For other possible external baselines (non-LLM classifiers), their performance on intellectual property auditing tasks is extremely poor. This is primarily because merchants on e-commerce platforms have developed sophisticated adversarial strategies that effectively conceal infringement information. Moreover, unimodal classifiers such as logo detectors can only provide symbolic outputs that require manually designed rules to reach a final decision, which introduces significant noise and bias. Therefore, accuracy-based evaluation is the most reliable and comparable metric across all models under study.
>
> 2. The arguments presented in Section 4.2
> For all methods and models, we prioritize maximizing processing throughput before considering memory efficiency. Section 4.2 primarily aims to illustrate that our proposed framework achieves both higher inference speed and lower memory footprint than competing approaches under equivalent throughput constraints.
>
> 3. Missing reference on line 156
> We have added the missing reference.
>
> 4. Missing character on line 93
> The missing symbol was a “greater than” sign, which has now been corrected.
>
> 5. Unclear notation in Equations (2) and (3)
> \( b \) denotes a brand within the controlled brand set, and \( B_t \) represents the set of brands detected from a given product description \( t \) using a BERT-based Named Entity Recognition (NER). We have updated the equations.
>
> 6. LogoBank
> LogoBank refers to our internally maintained retrieval database, which serves as a curated embedding repository of brand logos. It is not an existing external work but rather an internally developed resource designed for efficient similarity matching. The database construction process is straightforward and reproducible following the description provided in the paper.
>
> 7. Exemption expert
> Indeed, the exemption expert can be regarded as a rule-based decision tree. However, in practice, exemption policies in e-commerce vary significantly across countries and regions, making it impossible to implement a single deterministic if/else logic. The model must account for partial exemptions that depend on product categories, co-branded authorizations, and jurisdictional exceptions.
>
> 8. Law expert
> The law expert is a domain-specialized model fine-tuned on a dataset of QA-style pairs derived from decomposed infringement rules. Given the outputs of the preceding experts, it predicts the most probable type of violation based on brand status and field information, then retrieves the corresponding legal rationale.
>
> 9. \( D_{rev} \) in audit reversal data augmentation
> The generation of \( D_{rev} \) is essentially based on logical negation. The comprehensive expert outputs \( D_{orig} \) only when all three information-providing experts and the law expert agree on a consistent decision. Otherwise, it outputs the opposite label. A subset of reversed samples is manually inspected to ensure the validity and quality of the augmented data.
>
> 10. Ablation study of excluding specific experts
> Removing any specific expert would result in the loss of crucial modality-specific information. Intellectual property violations frequently manifest in only one modality (image, text, or metadata). Randomly removing an expert would therefore distort the data distribution and undermine the representativeness of the evaluation.
>
> 11. IPR-Audit dataset
> We do not plan to publicly release the IPR-Audit dataset due to data anonymization and intellectual property protection considerations. The dataset contains real trademarks and reflects platform-specific compliance logic. Releasing it would risk exposing sensitive audit rules and potentially aid malicious actors in evading detection. Such disclosure would raise serious legal and commercial concerns; therefore, the dataset will remain internal, though we may release anonymized summaries and model weights to support reproducibility.
>
> Once again, thank you very much for your time and thoughtful feedback. If these clarifications address your concerns, we would be deeply grateful if you could consider raising your evaluation. Should you have any further questions, we would be delighted to discuss them.

---

### Author Response · Authors · 2025-11-26
**We sincerely thank the Area Chair and all reviewers for the time, care, and expertise invested in evaluating our submission.**

We sincerely thank the Area Chair and all reviewers for the time, care, and expertise invested in evaluating our submission. Your detailed comments have been extremely helpful, and we learned a great deal from this discussion.

In response, we are actively re-considering (i) whether and how to disclose the dataset, and (ii) how to reorganize and clarify the method presentation to better address the concerns raised.

After careful consideration, we plan to withdraw the submission in one day. If there are any additional questions or clarifications you would like before then, we are happy to respond promptly.

Many thanks to all the reviewers for their comments.

---

> ### Comment · Reviewer_wsLt · 2025-11-26
> **Thanks**
>
> I would like to offer some personal suggestions regarding the quality of the manuscript:
>
> If privacy or copyright restrictions prevent the disclosure of the dataset, I suggest authors supplement it with fine-grained information such as its composition, features, statistics, and modalities in the appendix. Additionally, a brief description of how the dataset was collected and processed can be included, facilitating reader reproduction of your methods.
>
> Secondly, authors need to focus on improving the writing, such as reorganizing the paper structure and adding more formalized and clearer figures and tables.
>
> Finally, authors should emphasize the novelty of their methodology, requiring them not only to clearly articulate the core design but also to clearly outline the key challenges in the current field. Motivation is always paramount.

---

### Note · Authors · 2025-11-27

**Comment:**

We sincerely thank the Area Chair and all reviewers for the time, care, and expertise invested in evaluating our submission. Your detailed comments have been extremely helpful, and we learned a great deal from this discussion.

Unfortunately, the data is subject to commercial confidentiality constraints, so we are not able to publicly release it in full (even under an anonymous setting). Given this constraint, we are re-considering whether it is appropriate to continue with an ICLR submission, since we may not be able to meet the community’s expected standard for independent reproduction and verification.

After careful consideration, we decide to withdraw the submission.

Many thanks to all the reviewers for their comments.

**Withdrawal Confirmation:**

I have read and agree with the venue's withdrawal policy on behalf of myself and my co-authors.